# Development and Characterization of Emulsion Gels with Pine Nut Oil, Inulin, and Whey Proteins for Reduced-Fat Meat Products

**DOI:** 10.3390/foods14060962

**Published:** 2025-03-12

**Authors:** Berik Idyryshev, Alibek Muratbayev, Marzhan Tashybayeva, Assem Spanova, Shyngys Amirkhanov, Assel Serikova, Zhaksylyk Serikov, Laila Bakirova, Madina Jumazhanova, Aigerim Bepeyeva

**Affiliations:** Department of “Food Technologies”, Shakarim University, 20A Glinki Street, Semey 071412, Kazakhstan; b_1991@mail.ru (B.I.); marzhan06081990@gmail.com (M.T.); asem0678@gmail.com (A.S.); shyngys_a@inbox.ru (S.A.); instantly@list.ru (A.S.); zhak.7575@gmail.com (Z.S.); bakirova2010@mail.ru (L.B.); madina.omarova.89@mail.ru (M.J.); bepeyeva1987@gmail.com (A.B.)

**Keywords:** emulsion gel, inulin, pine nut oil, trans-isomers, polysaccharide, carrageenan

## Abstract

An emulsion gel was developed to replace animal fats in meat products while preserving desirable sensory and structural attributes. The gel was prepared by emulsifying pine nut oil and sunflower oil with whey protein concentrate (WPC) and polysaccharides (inulin and carrageenan). Process parameters, including the inulin-to-water ratio, homogenization speed, and temperature, were optimized to achieve stable gels exhibiting high water- and fat-binding capacities. Scanning electron micrographs revealed a cohesive network containing uniformly dispersed lipid droplets, with carrageenan promoting a denser matrix. Chemical assessments demonstrated a notably lower saturated fatty acid content (10.85%) and only 0.179% trans-isomers, alongside an elevated proportion (71.17%) of polyunsaturated fatty acids. This fatty acid profile suggests potential cardiovascular health benefits compared with conventional animal fats. Texture analyses showed that carrageenan increased gel strength and hardness; Experiment 4 recorded values of 15.87 N and 279.62 N, respectively. Incorporation of WPC at moderate levels (3–4%) further enhanced the yield stress, reflecting a robust protein–polysaccharide network. These findings indicate that the developed emulsion gel offers a viable alternative to animal fats in meat products, combining superior nutritional attributes with acceptable textural properties. The substantial polyunsaturated fatty acid content and minimal trans-isomers, coupled with the gel’s mechanical stability, support the feasibility of creating reduced-fat, functional formulations that align with consumer demands for healthier alternatives.

## 1. Introduction

Modern approaches to nutrition worldwide are increasingly focused on the development and consumption of foods that promote health. Particular attention is given to so-called functional foods, which have the potential to improve the body’s condition and reduce the risk of various diseases. Among these are products enriched with polyunsaturated fatty acids (PUFAs), which contribute to lowering the risk of cardiovascular and oncological diseases, normalizing cholesterol levels, strengthening the immune system, and increasing resistance to infections [1,2,3].

Nevertheless, many countries, including Kazakhstan, are experiencing an alarming rise in obesity levels. In 2008, 54,228 cases of obesity were recorded in the country, with higher prevalence among women (21.6%) than men (11%) [4]. Experts attribute this to the increased availability of high-calorie and fatty foods, as well as a decline in physical activity among the population [5]. At the same time, many traditional meat products (such as sausages) contain fat levels as high as 20–30% [6,7], which often leads to excessive consumption of calories and saturated fats.

Considering the recommendations of the World Health Organization (WHO), which defines the optimal ratio of fatty acids in the diet (no more than 10% saturated, 6–10% polyunsaturated, 10–15% monounsaturated, and trans fats not exceeding 1%) and cholesterol intake limit (300 mg per day) [8], it becomes evident that reducing the content of saturated fats in food products is necessary. One effective approach to achieving this goal is the partial or complete replacement of animal fats with PUFA-enriched components, particularly vegetable oils. However, the simple addition of liquid oil negatively affects the texture of meat products as liquid vegetable fats do not provide the same elastic structure at room temperature as animal fats [9,10].

In this context, emulsion gels are gaining popularity as they allow for the “imitation” of the physical properties of animal fats. These matrices are characterized by high water-holding capacity, ease of transportation, and improved oxidative stability [11,12]. Such emulsion structures can serve as sources of bioactive and phenolic compounds, as demonstrated in the study [13], which introduced an emulsion gel based on olive oil and soy protein, thereby providing a high content of beneficial components in sausages. Additional research has employed various plant- and animal-based by-products (pork skin, plant fibers, berry extracts) to develop similar emulsion gels, revealing that these innovative systems can enhance the oxidative stability of meat products and improve their nutritional profile [14,15].

Building on the demonstrated advantages of plant-derived emulsion gels, this study turns to pine nut oil—a resource abundant in polyunsaturated fatty acids from Eastern Kazakhstan—as a promising gel substitute for animal fats, aiming to maintain both nutritional quality and palatability in meat products. Pine nut oil is characterized by high digestibility and is notably rich in polyunsaturated fatty acids (PUFAs), primarily linoleic (up to 59.1%) and γ-linolenic acids (14.8–24.4%), which confer significant cardiovascular benefits [16]. Its substantial vitamin E content (~55 mg/100 g) greatly exceeds levels found in olive (14 mg/100g) and sunflower (41 mg/100g) oils, conferring pronounced antioxidant activity and resistance to oxidation. Additionally, pine nut oil provides essential fatty acids (collectively termed vitamin F), which crucial in cardiovascular disease prevention [17,18]. Due to its excellent nutritional profile and stability, pine nut oil is increasingly utilized in food technology, pharmaceuticals, and cosmetics.

The proposed technology involves the formulation of an emulsion gel in which vegetable oil (pine nut oil) is structured using a polysaccharide (inulin) and additional stabilizing components to form a gel-like structure. This approach aims to enhance the organoleptic properties and nutritional value of products with reduced animal fat content.

Hydrocolloids (polysaccharides), including inulin, are widely used in the formulation of such recipes. They can stabilize emulsions and form a plastic structure similar to fat [19,20,21]. Inulin, found in chicory, Jerusalem artichoke, and several other plants, can form a gel-like precipitate with a very fine crystalline structure that retains water under certain conditions [22,23]. The resulting gel is nearly tasteless and has a texture similar to animal fat. In the food industry, inulin is increasingly used as a functional and technological ingredient: it serves as a dietary fiber while also acting as an emulsion stabilizer or even a fat replacer [24,25].

The goal of this research is to develop pine nut oil-based emulsion gels stabilized by inulin, carrageenan, and whey protein concentrate (WPC) as healthier alternatives to animal fats in meat products, specifically aiming to enhance structural stability, improve textural properties, and achieve a nutritionally favorable fatty acid profile.

## 2. Materials and Methods

### 2.1. Sample

In this study, all materials were acquired from recognized commercial suppliers to ensure consistency and traceability. Inulin Beneo (Beneo Orafti, Belgium), exhibiting a degree of polymerization exceeding 23, served as the primary polysaccharide for gel formation. Pine nut oil (Pinus sibirica) acquired from LLP «Arshaty» (Katon-Karagay, Kazakhstan) and sunflower oil supplied by Eurasian Foods Corporation (Almaty, Kazakhstan) were used as lipid sources.

Carrageenan was purchased from LLP “Khim-Baza” (Almaty, Kazakhstan), providing additional structural support for emulsion gel formation. Carrageenan (E-407) is a natural food additive derived from red seaweeds and is recognized as safe for use by international regulatory authorities, including the Codex Alimentarius Commission and the Technical Regulation TR TS 029/2012, which confirms its approved safety status and defines specific usage guidelines [26,27].

### 2.2. Preparation of Inulin–Water Dispersion

An inulin–water dispersion was first prepared to serve as the primary gel-forming component. The required mass of inulin was weighed using an analytical balance (±0.001 g accuracy) and transferred into a clean, dry vessel. Deionized water was heated to a temperature of approximately 18–20 °C for improved solubility, and then gradually added to the inulin under continuous stirring. Mechanical agitation was maintained using a high-speed homogenizer set at 1000–1500 rpm for 10–15 min. This mixing duration ensured uniform hydration of the inulin particles, promoting the partial dissolution and dispersion necessary for subsequent gel formation. After this initial mixing phase, the inulin–water system was allowed to rest for an additional 5 min to facilitate further swelling of the inulin particles.

### 2.3. Incorporation of Carrageenan

Once the inulin–water dispersion reached a stable consistency, carrageenan was introduced to improve gel strength and thermal stability. A predetermined quantity of carrageenan was weighed, and the powder was gently sprinkled into the inulin dispersion. Stirring resumed at the same homogenizer speed to prevent local clustering of carrageenan. Mixing continued for 5–7 min, ensuring that the carrageenan dissolved and integrated uniformly throughout the inulin–water matrix. The temperature was maintained at 18–20 °C to avoid premature gelation, which can occur if carrageenan solutions are subjected to higher temperatures followed by rapid cooling. By carefully monitoring the temperature and mixing times, a cohesive polysaccharide network was formed, laying the foundation for a stable gel.

### 2.4. Addition of Oil and WPC

Pine nut oil and sunflower oil were incorporated as the lipid phase to create the emulsion gel. The oil mixture was measured according to the experimental formulation and slowly introduced into the hydrated inulin–carrageenan system. To achieve a uniform emulsion, the homogenizer speed was increased stepwise up to 1500 rpm, and emulsification continued for 15–18 min. During this period, the formation of small, uniformly distributed oil droplets was facilitated by the presence of the polysaccharides, which acted as emulsifiers or stabilizing agents. The temperature was kept below 30 °C to minimize thermal fluctuations that could disrupt the emulsion. Care was taken to avoid foam generation by positioning the homogenizer head slightly below the surface of the mixture.

After establishing a stable dispersion—achieved by homogenizing the polysaccharide–oil–water mixture (e.g., at 1500 rpm for 10–15 min)—the required amount of WPC was introduced gradually into the mixture over 1–2 min. Stirring continued at the same homogenization speed for an additional 2–3 min to ensure uniform distribution of WPC, allowing the proteins to fully integrate into the polysaccharide and lipid phases.

### 2.5. Final Composition Formation and Storage

Upon completion of emulsification, the resulting mixture was immediately transferred to a sterile container. The composition was then placed in a refrigerated environment (0–4 °C) for a minimum of 6–8 h. This cold storage period allowed the inulin–carrageenan network to fully develop, immobilizing the lipid droplets within a three-dimensional matrix. The final emulsion gel (Figure 1) was evaluated for structural, rheological, and compositional properties before further incorporation into meat-based or other low-fat food products. All equipment in contact with the material was thoroughly sanitized between batches to ensure consistent results and prevent cross-contamination.

### 2.6. Determination of Water-, Fat-Absorbing Capacity

The water absorption capacity of inulin was determined using a stainless steel mesh beaker (height 80 mm, diameter of mesh holes 1.5 mm, number of holes per 1 cm—10–20). The bottom and walls of the beaker were covered with filter paper to avoid loss of fine particles. The beaker was wetted with water, then the water was drained for 20 min and the beaker was weighed. A total of 2 g of inulin was placed in it, after which the beaker with the suspension was immersed in room temperature water for 20 min. After draining for 20 min, the outer walls and bottom of the beaker were wiped with filter paper, weighed, and calculated [28].

The fat-absorbing capacity of inulin was determined using a stainless steel mesh beaker (height 80 mm, diameter of mesh holes 1.5 mm, number of holes per 1 cm—10–20). The bottom and walls of the beaker were covered with filter paper to avoid loss of fine particles. The beaker was wetted with vegetable oil, then within 20 min, the vegetable oil was drained off and the beaker was weighed. A total of 2 g of inulin was placed in it, after which the beaker with the suspension was immersed in room temperature vegetable oil for 20 min. After draining for 20 min, the outer walls and bottom of the beaker were wiped with filter paper, weighed, and calculated [28].

### 2.7. Determination of Critical Gelation Concentration

The critical gelation concentration (CGC) was determined by preparing a series of inulin suspensions of different concentrations (from 10 to 17%) with a step of 1%. The preparation of solutions was carried out using a homogenizer, and the working bodies stirred at *n* = 1000 rpm for 15 min. The obtained viscous solutions were poured into test tubes and incubated at 0–4 °C for 6–8 h. After this time, lead balls with a diameter of 4 mm and an average mass of 0.53 g were placed on the gel surface and incubated for 2 h at 0–4 °C. The minimum inulin concentration corresponding to the sample in which the gel was not destroyed under the influence of a lead ball was taken as the CCG [29].

### 2.8. Determination of Gel Strength and Hardness

Gel strength was determined through Texture Profile Analysis (TPA) using a TA.XT Plus texture analyzer (Stable Micro Systems, Godalming, UK). Cylindrical samples of uniform dimensions were prepared to ensure consistent testing conditions. The analysis was performed at a constant test speed of 1.0 mm/s with a cylinder probe (e.g., P/0.5). Each sample was subjected to a compression test, during which the force required to deform or rupture the gel was recorded. This force was taken as the measure of gel strength. In complementary compression tests, the samples were compressed to a specified percentage (e.g., 50%) of their initial height, and the force resisting this compression was used to further characterize gel strength.

Hardness was assessed via TPA under conditions similar to those employed for gel strength. The texture analyzer settings included a pre-test speed of 1.0 mm/s, a test speed of 0.5 mm/s, a post-test speed of 1.0 mm/s, a trigger force of 5.0 g, and a 50% compression ratio. Samples were compressed twice, simulating a chewing action with a brief interval between compressions to permit the measurement of additional parameters, such as springiness, cohesiveness, gumminess, chewiness, and resilience. Hardness was defined as the maximum force observed during the first compression cycle [30].

### 2.9. Determination of Proximate Composition and pH

The proximate composition was determined using standard AOAC methods [31]. Protein was measured by the Kjeldahl method using a nitrogen-to-protein conversion factor of 6.25. Fat was determined by Soxhlet extraction with petroleum ether as the solvent. Ash was measured by incinerating the sample at 525 °C. Moisture was obtained by oven-drying the sample until a constant weight was reached.

The total carbohydrate content was calculated by Equation (1):Total carbohydrate content = 100 − (protein + lipids + ash + moisture)%(1)

Hydrogen ion concentration (pH) was determined by potentiometric method using PHS-3D-03 (Shanghai San-Xin Instrumentation Inc., Shanghai, China) according to GOST R 51478-99108 [32].

### 2.10. Microstructural Analysis

The microstructure of the emulsion gels was investigated using a low-vacuum scanning electron microscope (JSM-6390LV, JEOL, Tokyo, Japan). Samples were placed in the microscope chamber without additional conductive coating, and an accelerating voltage of 15 kV was applied to the electron beam. Observations were performed at a magnification of 100× to assess the overall morphology and pore distribution within the gel matrix [33].

### 2.11. Determination of Fatty Acid Composition

The fatty acid composition of the product was determined according to GOST R 55483-2013 by extraction of lipids from gel samples by chloroform/methanol extraction according to Folch’s method [34]. The purity of isolated lipids was checked by thin-layer chromatography. Fatty acid composition was determined on a HP 6890 gas chromatograph manufactured by “Hewlett Packard” with a flame ionization detector and HP-Innowax column 30 m × 0.25 mm × 0.25 mkm (“Agilent Technologies”, Santa Clara, CA, USA).

### 2.12. Statistics

Statistical processing of data was performed using standard Microsoft Excel and Statistica program packages. The studies were conducted in 3 parallels. Basically, the data are presented in the format of arithmetic mean ± standard deviation. Deviation of more than 5% was taken as statistically insignificant.

## 3. Results and Discussion

### 3.1. Inulin: Water Hydromodule

In the present work, the feasibility of using inulin as a structuring ingredient in an emulsion gel was investigated. The research addressed the functional and technological properties of inulin when dispersed and gelled in water under varying process conditions. The study also identified the optimal inulin-to-water ratio (hydromodulus) and mixing parameters required to achieve the desired gel strength. The results concerning inulin’s water- and fat-absorbing capacity indicated values of 189.0 ± 11.1% and 114.0 ± 12.2%, respectively (Table 1). Such high absorption capacities favor reduced-fat formulations by ensuring adequate moisture retention and contributing to a creamy texture.

Inulin exhibits strong moisture-retention properties primarily due to its ability to form a three-dimensional network structure through intermolecular hydrogen bonding. Its long-chain polymer structure traps water molecules efficiently, thus enhancing the moisture retention and textural quality in emulsion gels and significantly improving the juiciness and sensory characteristics of reformulated meat products [35,36]. This finding aligns with existing literature that notes inulin’s ability to form stable, fat-like gels in low-fat products [37,38]. Tests on the gel-forming ability revealed that a 10% inulin concentration was sufficient to initiate gelation, while 15% served as the critical gelation concentration (CGC), defined by the capacity of the gel to support a standard load without collapsing. This threshold of 15% was thus deemed crucial for achieving a stable network capable of improving the textural attributes of meat products.

A series of gels prepared with inulin-to-water ratios ranging from 1:5.5 to 1:3 underwent yield stress measurements to quantify gel strength (Figure 2). At the most dilute ratio (1:5.5), the yield stress was 480 Pa, increasing steadily up to 1851 Pa at the most concentrated ratio (1:3). These data illustrate that reducing the hydromodulus, or equivalently increasing inulin content, promotes the formation of a denser network and higher yield stress. Given that spreadable or pâté-type products typically require a yield stress between 800 and 1200 Pa [39,40], ratios of 1:4 to 1:3.7 (with respective yield stress values ranging from 798 Pa to 1256 Pa) were identified as optimal.

The influence of mixing parameters was also explored at an inulin concentration of 20%. Prolonged mechanical dispersion for 5, 10, and 15 min demonstrated a progressive increase in yield stress from 576 Pa at the shorter mixing time to 794 Pa at the longer one (Figure 3). This enhancement of gel strength is attributable to more complete hydration of inulin and improved dispersion of its particles in the aqueous phase. Varying the homogenizer rotation speed from 500 to 3000 rpm further revealed that speeds up to approximately 1500 rpm encouraged a significant increase (22–24%) in yield stress, whereas beyond 1500 rpm, further acceleration contributed minimal improvements. From a practical standpoint, a rotation speed of 1500 rpm for about 10 min effectively balances energy consumption with gel strength development.

These findings have practical implications for creating emulsion gels that reduce overall fat content in meat products while retaining desirable sensory properties, such as mouthfeel, texture, and juiciness. By establishing a CGC of at least 15% and optimizing the hydromodulus (1:4 to 1:3.7) and mixing parameters (1500 rpm, 10 min), inulin-based gels can meet the mechanical requirements of many meat applications, including those traditionally reliant on animal fat. Inulin also exhibits a strong capacity for water retention, contributing to the moist, creamy characteristics necessary in sausages, pâtés, and related formulations [41]. When combined with other components, such as whey protein concentrate, pine nut oil, and carrageenan, inulin can form a robust emulsion-like matrix that mimics the functionality of animal fat and aligns with consumer interest in healthier products featuring lower saturated fat and higher dietary fiber [42,43].

Inulin’s excellent water- and fat-absorbing capacities, along with its robust gel-forming properties, allow for partial or total replacement of animal fats in meat formulations. Previous studies report that native inulin only gels above ~30% (*w*/*w*) solids, while long-chain inulin can gel around 20–40% (*w*/*w*) [44]. Chicory inulin dispersions remain fluid at 25% but transition to a gel-like solid at ≥27.5% (*w*/*w*) [45]. These concentrations are in line with our optimized inulin-to-water ratio, which produced a stable gel structure. In this study, the inulin concentration of approximately 20% (hydromodulus 1:4) combined with moderate homogenizer speeds (about 1500 rpm) and mixing durations of 10 min produces a stable gel structure suitable for soft-textured meat products.

### 3.2. Determination of the Optimal Amount of Oil and Milk Whey Protein Concentrate (WPC) in the Emulsion Gels

This stage of the study focused on designing an emulsion gel by combining polysaccharides (inulin or a blend of inulin and carrageenan) with vegetable oils (Table 2). Pine nut oil produced by cold pressing was used as the principal fat component, supplemented by sunflower oil at varying levels.

Inulin demonstrates the ability to form complexes with a range of hydrocolloids, such as starch, gelatin, pectin, and gum [46]. These interactions can lead to increases or decreases in viscosity and can modulate gel properties, including brittleness, elasticity, and strength. By contrast, inulin-based gels exhibit reduced strength upon heating, likely because hydrogen bonds are broken and partial hydrolysis to fructose occurs; the gel structure is not fully restored upon cooling. Carrageenan, however, forms thermally reversible gels when heated to approximately 70 °C and then cooled below 40 °C [47]. The complementary characteristics of inulin and carrageenan suggest that their combined use may result in synergistic gelling, yielding stable textures with favorable spreadability.

A high-speed emulsification stage at 1500 rpm for 15 min was central to the mixing process, after which oil was gradually added over a total processing time of about 15–18 min. The resulting emulsions were then maintained at 0–4 °C for 6–8 h to allow for complete structure formation. As summarized in Table 2, the proportion of pine nut oil ranged from 10% to 35%, while inulin or inulin–carrageenan mixture served as the stabilizing agent. Compositions based solely on inulin displayed partial phase separation, with up to 70% of the oil separating at higher oil levels. In contrast, substituting part of the inulin with carrageenan prevented this separation and ensured full incorporation of the oil within the emulsion.

Whey protein concentrate (WPC) was incorporated into each emulsion gel sample by first preparing the base emulsion of inulin, carrageenan (when applicable), water, and oil according to the specific formulation (Table 2). This step-by-step protocol was repeated for each experiment, with WPC contents varying from 0% to 6%.

Carrageenan and inulin interact synergistically through electrostatic and hydrogen bonding. Carrageenan, bearing negatively charged sulfate groups, interacts strongly with positively charged regions on whey proteins, stabilizing the gel structure. Inulin’s long-chain polymers form crystalline aggregates, further strengthening this network. Together, these interactions create a robust, stable emulsion gel, enhancing water retention, texture, and structural integrity in meat products [48,49,50].

Because milk whey proteins are acknowledged as efficient emulsifiers, introducing WPC was expected to enhance oil entrapment by creating a continuous proteinaceous shell around the fat droplets [51,52]. To evaluate this effect, experimental samples were produced by incrementally adding between 2.5% and 12.5% WPC to aqueous dispersions of inulin or inulin–carrageenan, followed by the introduction of pine nut and sunflower oils under high-speed homogenization at 1500 rpm for a total of 15–18 min. The resultant mixtures were held at 0–4 °C for 6–8 h, allowing the structure to fully develop.

A clear increase in yield stress was observed across all emulsion gel samples (Experiments 1–6) with rising concentrations of whey protein concentrate (WPC). At 0% WPC, the yield stress ranged from 612 Pa (Experiment 1) to 944 Pa (Experiment 6). Raising WPC content to 6% led to substantial increases, with Experiment 6 reaching the highest yield stress value of 3362 Pa (Figure 4). These observations suggest that WPC contributes to the formation of a stronger three-dimensional network, likely through additional protein–polysaccharide interactions and improved emulsifying capacity.

Samples containing carrageenan (Experiments 2, 4, and 6) consistently exhibited higher yield stress values than formulations lacking carrageenan (Experiments 1, 3, and 5) at comparable WPC levels. This difference can be attributed to the synergistic effect between carrageenan and WPC: negatively charged sulfate groups on the carrageenan chain interact with the positively charged domains of whey proteins, thereby generating a denser gel matrix [53]. Notably, Experiments 5 and 6, which incorporate higher oil concentrations, do not meet the target textural parameters: the marked rise in yield stress at elevated WPC levels indicates excessive elasticity. Such gels may no longer maintain the desirable balance between spreadability and stability, rendering them less suitable for typical meat product applications. Consequently, while higher WPC and carrageenan contents improve yield stress, care must be taken to avoid excessively rigid gels that deviate from the intended functional properties. Experiment 3 and Experiment 4 demonstrated the most desirable balance between yield stress and visual appearance when 3–4% WPC was added. At these concentrations, the emulsion gels displayed a stable network structure without becoming excessively rigid and showed minimal visible phase separation. Based on these findings, Experiments 3 and 4 were selected for subsequent investigations.

Prior research on cold-set whey protein gels found that increasing protein from ~4% to 6% significantly raises the storage modulus and breaking force of the gel matrix [54]. Moakes et al. reported that higher oil volume fractions in whey protein fluid gels led to pseudo-solid behavior with an apparent yield stress due to droplet crowding increasing the effective phase volume [55]. This is attributed to a denser protein network that better distributes stress. Using a moderate WPC level (≈3–4%) in this study aligns with these findings, as doing so was sufficient to form a robust protein-polysaccharide network without compromising texture.

### 3.3. The Chemical Compositions of Emulsion Gels

The chemical compositions of these two formulations showed that Experiment 3 had a slightly higher moisture content of 57.65% and lower protein and fat contents, at 3.11% and 19.82%, respectively, compared with Experiment 4. The pH values were 5.13 for Experiment 3 and 5.28 for Experiment 4. Experiment 4 exhibited more protein (3.85%) and fat (20.79%), reflecting the combined impact of including carrageenan and a higher proportion of oil (Table 3).

Our emulsion gel is highly hydrated (water content <50%), which is an advantage in meat applications. This extra water improves juiciness and reduces calorie density. Indeed, replacing animal fat with water–polymer gels in meat patties significantly increases retained moisture compared to full-fat controls. Similar fat-replacer gels have been shown to elevate the moisture content of final products by ~9–16% as the gel introduces additional water that is bound in the matrix [56]. The protein content in emulsion gel comes mainly from WPC. While modest in absolute terms, this protein improves the gel’s nutritional quality (adding a source of amino acids) and functional properties. Comparable fat-replacer gels often use ~2–5% protein (whey, soy, or egg proteins) to stabilize emulsions [57].

Additional assessments of water-holding capacity (WHC) and fat-holding capacity (FHC) under varying homogenizer speeds from 500 to 3000 rpm demonstrated that both WHC and FHC peaked at moderate speeds (1500–2000 rpm) (Figure 5 and Figure 6). At speeds above this range, these capacities declined, likely due to excessive shear forces causing partial disruption of the gel network. Notably, Experiment 4 consistently achieved superior fat and water retention, reinforcing the premise that carrageenan enhances the emulsifying potential of inulin and WPC.

### 3.4. Gel Strength and Hardness of Emulsion Gel Samples

In Experiment 3, the gel strength was measured at 13.56 N, while hardness reached 227.53 N. By contrast, Experiment 4 exhibited elevated values of 15.87 N and 279.62 N, respectively. The increase in both parameters can be attributed primarily to the inclusion of carrageenan, which forms thermally reversible gels and interacts synergistically with inulin to create a denser, more cohesive hydrocolloid network. This enhanced network structure provides greater resistance to deformation, thereby increasing gel strength and hardness.

Other studies have measured hardness in emulsion gels and found trends that align with our observations. For instance, a whey protein emulsion gel without added polysaccharide showed a low hardness around 1.3 N, indicating a soft gel. When hydrocolloids like basil seed gum or xanthan were incorporated, hardness more than doubled (up to 3 N) as the network strengthened [58]. This demonstrates that polysaccharides greatly reinforce the gel matrix. Our results with carrageenan mirror that effect but on a larger scale; at sufficient levels, κ-carrageenan forms its own gel network, which synergistically boosts gel firmness. In a recent study using pea protein and κ-carrageenan, increasing carrageenan concentration by up to 1.5% raised gel hardness 75-fold [59].

Subsequent structural and mechanical measurements revealed that Experiment 4 possessed greater strength and stickiness than formulations without carrageenan. The synergy between inulin and carrageenan contributed to an approximately 13–15% increase in gel strength relative to recipes based only on inulin. This outcome highlights the feasibility of using these polysaccharides in tandem to regulate and bolster the structural and mechanical properties of emulsion-type products, an attribute of particular importance for meat product applications. The production steps involved hydrating and homogenizing the polysaccharides, gradually adding WPC, introducing pine nut and sunflower oils, and maintaining the final emulsion at low temperature for several hours to promote gel network consolidation. These findings collectively indicate that moderate WPC concentrations, when used alongside inulin and carrageenan, considerably improve an emulsion gel’s strength, stability, and emulsification. Such advantages underscore the suitability of these systems for producing emulsified meat items, in which texture, moisture management, and fat retention constitute key quality attributes.

### 3.5. Microstructure of Emulsion Gels

In the micrograph of Experiment 3 (Figure 7), numerous spherical droplets measuring between approximately 10 and 25 µm in diameter are visible, suggesting a well-formed emulsion network. These droplets appear interconnected by thin, thread-like structures, which likely represent polysaccharide–protein filaments bridging adjacent oil droplets. The background surface appears relatively homogeneous, though local irregularities—manifested as discontinuities—indicate zones of either incomplete coalescence or natural heterogeneities in the matrix. Black voids observed among the droplets reflect areas devoid of solid or liquid phases, possibly indicating entrapped air pockets or localized shrinkage upon sample preparation. Variations in brightness and contrast, such as the brighter protrusions and darker depressions, correspond to topographical differences: more elevated regions scatter more electrons and thus appear brighter, while recessed or hollow areas appear darker.

In Experiment 4, the scanning electron micrograph shows relatively few spherical droplets and a predominantly planar surface interspersed with dark discontinuities representing pores or voids. Thin, thread-like streaks traverse the gel surface, and certain regions display oval or trapezoidal fragments. The incorporation of carrageenan appears to have influenced the matrix, reducing the presence of distinct droplets while promoting a more uniformly contoured gel structure with intermittent porous zones. This morphology suggests a tighter polysaccharide network, wherein carrageenan’s gelling properties likely contribute to fewer and more widely dispersed lipid droplets, thereby reinforcing the cohesive organization of the protein–polysaccharide matrix.

These morphological features support the presence of a cohesive, gel-like network wherein proteins and polysaccharides stabilize the oil phase, giving rise to a structured matrix with discreet and moderately uniform droplet size. Such an organization is essential for the functional properties of emulsion gels, including firmness, spreadability, and resistance to phase separation.

### 3.6. The Fatty Acid Profiles and Trans-Isomer Content

A comparative analysis of the fatty acid profiles and trans-isomer content of the newly developed emulsion gel (Experiment 4) and various animal fats reveals distinct nutritional advantages of the proposed formulation. The trans-isomer content in Experiment 4 is 0.179% of the total fat, which is notably lower than values measured in tail fat (2.908%), lamb fat (7.239%), and beef fat (5.024%). Such a pronounced difference in trans-isomer levels is generally favorable from a health standpoint, considering the recognized association between elevated trans-fat intake and adverse cardiovascular outcomes. The results also show that the developed composition’s trans-isomer content lies close to that of chicken fat (0.14%) and pork fat (0.26%), further underscoring the potential of this formulation as a lower-risk alternative to ruminant-derived fats (Table 4).

The fatty acid composition of Experiment 4 indicates that the saturated fatty acid (SFA) fraction is 10.85%, the monounsaturated fatty acid (MUFA) fraction is 17.98%, and the polyunsaturated fatty acid (PUFA) fraction is 71.17% of the total fat. This profile contrasts sharply with those of conventional animal fats. For example, lamb fat and tail fat exhibit saturated fatty acid levels of 56.24% and 50.42%, respectively, with correspondingly small PUFA percentages. In the newly developed composition, the extensive presence of PUFAs significantly exceeds those found in beef (9.34%), pork (7.32%), and chicken (15.25%) fats. Elevated PUFA levels, particularly in the omega-3 and omega-6 series, are known to contribute to beneficial physiological effects and reduced risk of cardiovascular disease [60]. A similarly low SFA fraction, exemplified by the 10.85% value in the developed composition, further enhances the lipid profile by mitigating potential risks associated with high saturated fat consumption.

A comparative assessment of the fatty acid profiles in the developed emulsion gel (Experiment 4) and various animal-derived fats reveals significant differences in both the total saturated fat content and the distribution of specific fatty acids. In the emulsion-type composition, the total share of saturated fatty acids (SFA) is 10.85%, with palmitic (6.81%) and stearic (3.74%) acids representing the most abundant SFAs, while numerous minor SFAs (e.g., butyric, caproic, lauric) remain undetected. By contrast, lamb and tail fats exhibit markedly higher SFA levels of 56.24% and 50.42%, respectively, and often contain additional medium-chain fatty acids such as capric or lauric acid in measurable amounts. Other sources, including beef and pork fat, present intermediate SFA contents of approximately 45.04% and 36.33%, respectively, highlighting a potentially greater risk of hypercholesterolemia and cardiovascular disease when consumed in excess. It is well recognized that high concentrations of SFAs can adversely affect human health by increasing low-density lipoprotein (LDL) cholesterol, thereby raising the risk of atherogenesis.

Analyses of monounsaturated fatty acids (MUFA) show that pork fat and beef fat contain 56.35% and 45.62% MUFA, respectively, largely in the form of oleic acid. The emulsion gel displays 17.98% MUFA, primarily oleic acid (17.73%), indicating a more balanced distribution that can help modulate plasma lipoproteins without substantially elevating LDL cholesterol. Such moderate MUFA levels, although below those of pork or beef fats, may still confer certain health benefits by promoting a favorable lipoprotein profile.

A noteworthy finding in the developed composition is the substantially higher content of polyunsaturated fatty acids (PUFA), amounting to 71.17% (Table 5). Linoleic acid dominates, representing 70.19% of total fatty acids, with minor contributions from linolenic (0.27%) and eicosatrienoic (0.56%) acids. This high proportion of PUFA far exceeds that found in other animal fats, including chicken (15.25%), horse (29.67%), lamb (4.89%), and beef (9.34%). A diet rich in PUFAs, particularly in n-3 and n-6 fatty acids, is associated with a reduced incidence of cardiovascular disorders and improved inflammatory markers, in part due to their role in modulating eicosanoid synthesis and cell membrane fluidity [61]. Consequently, substituting animal fats with the formulated emulsion gel may deliver advantages in reducing saturated fat intake and increasing biologically active unsaturated lipids. These data collectively underscore the nutritional and health-promoting potential of the emulsion gel with minimal SFA content and a high PUFA fraction.

These findings emphasize that replacing conventional animal fats with the proposed emulsion gel can substantially reduce both trans-isomers and saturated fatty acids while simultaneously increasing the proportion of PUFAs. From a nutritional standpoint, this shift aligns with dietary recommendations to improve lipid intake quality. The large fraction of unsaturated fatty acids also positively influences the physical properties of the emulsion gel, since lower levels of saturated and trans fatty acids typically yield more malleable and spreadable textures. Such attributes are essential in designing healthier sausage or meat products that maintain acceptable technological properties, including mouthfeel and emulsion stability. Overall, the markedly lower trans-isomer content and elevated PUFA levels in Experiment 4 demonstrate its promise as a superior alternative to traditional animal fats, both in terms of health-oriented profiles and functional characteristics relevant to the meat industry. The developed emulsion gels exhibit notable nutritional advantages due to their high polyunsaturated fatty acid (PUFA) content and significantly reduced saturated fats compared to animal fats. Elevated PUFA consumption is associated with improved cardiovascular health, reduced inflammation, and lower cholesterol levels. Additionally, the high dietary fiber content from inulin provides prebiotic benefits, potentially enhancing gut microbiota composition. Consequently, incorporating such gels into meat products represents a promising strategy for developing healthier food alternatives.

## 4. Conclusions

The present study demonstrates that an emulsion gel formulated from pine nut oil, inulin, carrageenan, and whey protein concentrate (WPC) constitutes a viable alternative to traditional animal fats in meat products, offering both nutritional and functional advantages. Optimizing the inulin-to-water ratio and homogenization parameters yielded robust gels with high water- and fat-binding capacities. Incorporation of WPC up to moderate levels (3–4%) further increased the yield stress of the emulsion gel, indicating enhanced structural integrity. This effect was particularly pronounced in samples containing carrageenan to a dense, cohesive network. Micrographs revealed differences in droplet size and arrangement depending on the formulation. Samples containing carrageenan showed fewer, more dispersed droplets within a compacted matrix, while inulin-only gels exhibited more prominent spherical droplets. Chemical assessments confirmed a favorable fatty acid profile, with only 10.85% saturated fatty acids and 0.179% trans-isomers, and a high content (71.17%) of polyunsaturated fatty acids. These features suggest potential health benefits related to lipid intake. Overall, the combination of enhanced textural properties, improved microstructural organization, and beneficial fatty acid composition underlines the suitability of this emulsion gel system for developing reduced-fat meat products that meet consumer demands for healthier options. Subsequent research will investigate how the emulsion influences meat product quality attributes, i.e., shelf life, consumer acceptance, flavor optimization, oxidative stability assessments.

## Figures and Tables

**Figure 1 foods-14-00962-f001:**
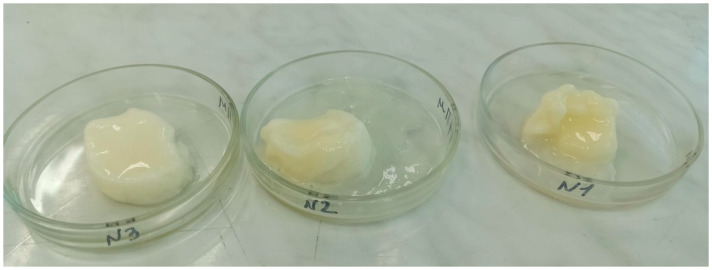
Appearance of emulsion gels.

**Figure 2 foods-14-00962-f002:**
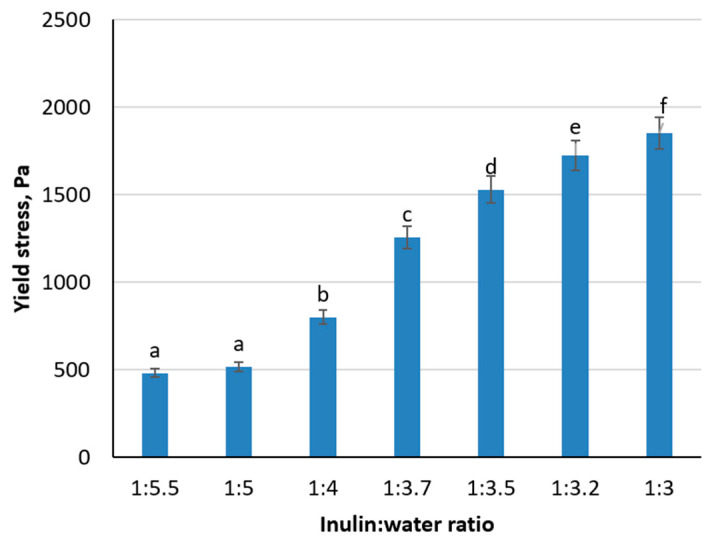
Influence of inulin: water hydromodule on the change of inulin gel yield stress index (Different letters above the bars indicate significant differences between samples, *p* < 0.05).

**Figure 3 foods-14-00962-f003:**
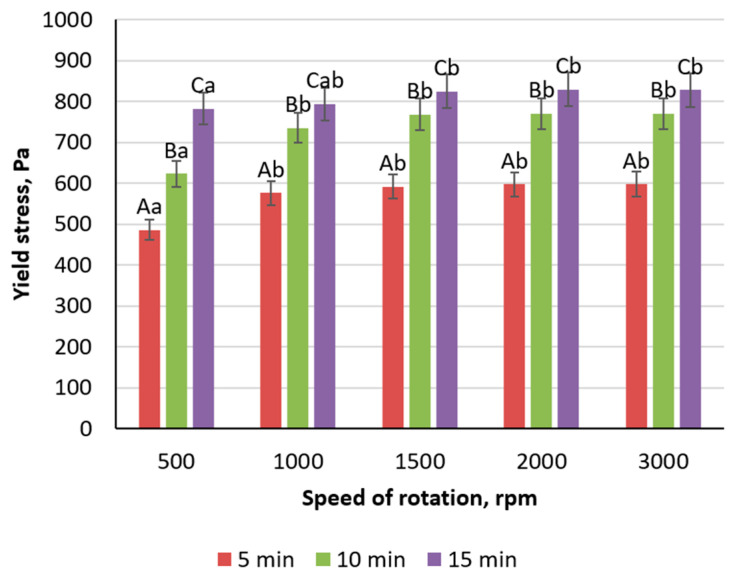
Effect of duration and speed of homogenizer rotation during mixing on the yield stress of gels prepared on the basis of inulin (Different lowercase letters (a,b) indicate statistically significant differences within the same time of mixing (*p* < 0.05). Different uppercase letters (A–C) indicate a significant difference within the same speed of rotation).

**Figure 4 foods-14-00962-f004:**
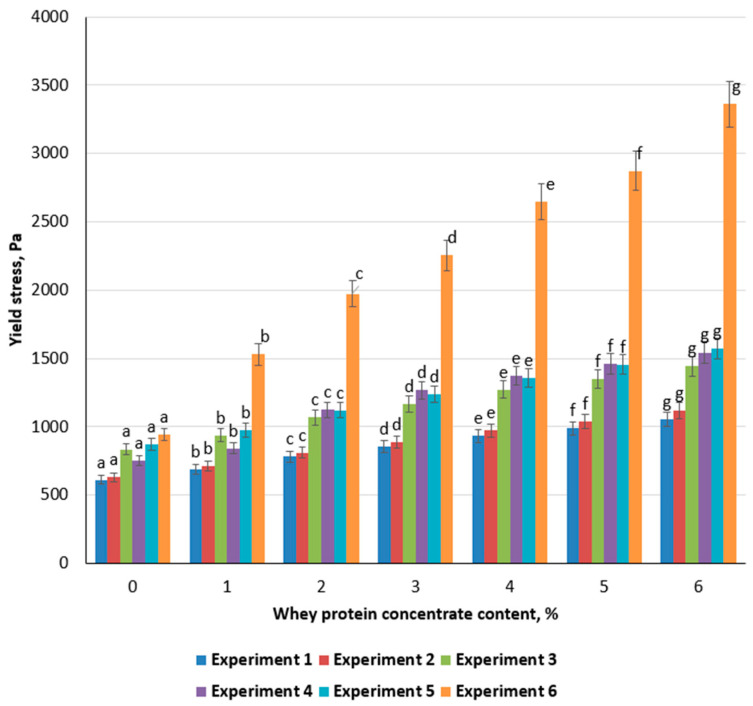
Effect of whey protein concentrate on the yield stress of emulsion gels. Different lowercase letters (a–g) indicate statistically significant differences within the sample but different concentration of WPC (*p* < 0.05).

**Figure 5 foods-14-00962-f005:**
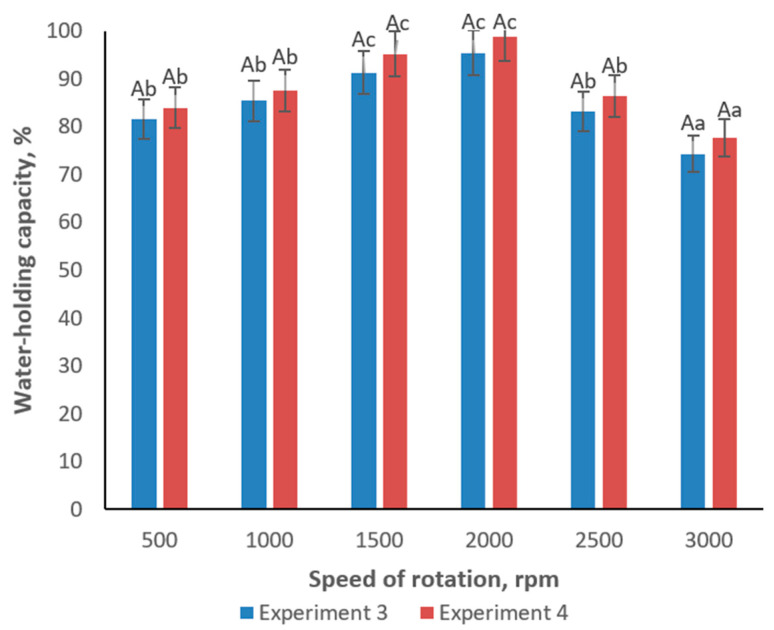
Change of water-holding capacity of emulsion gel samples in dependence of speed of rotation. Different lowercase letters (a–c) indicate statistically significant differences within the same sample but different speed of rotation (*p* < 0.05). Uppercase letter (A) within all samples indicates a non-significant difference (*p* > 0.05) within the same speed of rotation.

**Figure 6 foods-14-00962-f006:**
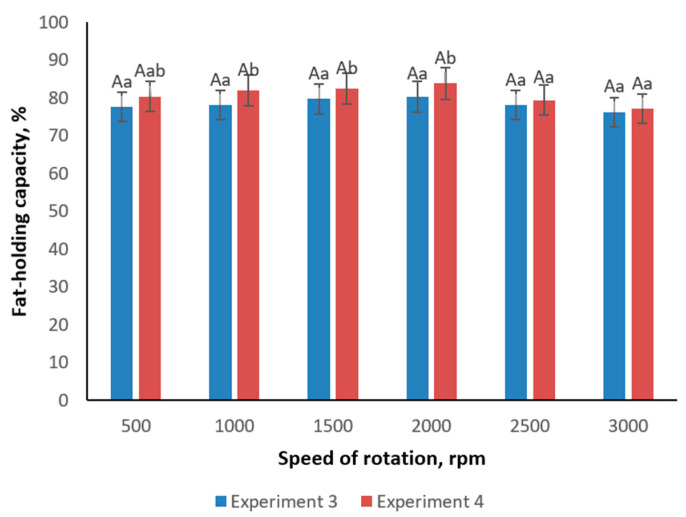
Change of fat-holding capacity of emulsion gel samples in dependence of speed of rotation. Different lowercase letters (a,b) indicate statistically significant differences within the same sample but different speed of rotation (*p* < 0.05). Uppercase letter (A) within all samples indicates a non-significant difference (*p* > 0.05) within the same speed of rotation.

**Figure 7 foods-14-00962-f007:**
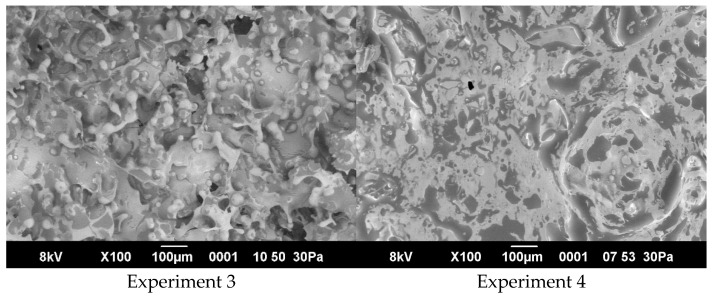
Microstructure of emulsion gels.

**Table 1 foods-14-00962-t001:** Functional and technological properties of inulin.

Indicator	Mean + SD
Fat absorption capacity, %	114.0 ± 12.2
Water absorption capacity, %	189.0 ± 11.1

**Table 2 foods-14-00962-t002:** Model samples of emulsion gels based on polysaccharides and vegetable oil.

Ingredients	Ingredients Consumption Per 100 g of Product, g					
	Experiment 1	Experiment 2	Experiment 3	Experiment 4	Experiment 5	Experiment 6
Pine nut oil	10	15	20	25	30	35
Sunflower oil	10	10	10	10	10	10
Inulin	20	18	20	18	20	20
WPC	1	2	3	4	5	6
Carrageenan	-	2	-	2	-	2
Water	59	53	47	41	35	27
Total	100	100	100	100	100	100

**Table 3 foods-14-00962-t003:** Chemical composition of emulsion gels.

Indicator	Experiment 3	Experiment 4
Water	57.65 ± 0.93	55.23 ± 0.72
Protein	3.11 ± 0.03	3.85 ± 0.06 *
Fat	19.82 ± 0.37	20.79 ± 0.22
Carbohydrate	18.21 ± 0.24	18.85 ± 0.36
Ash	1.21 ± 0.02	1.28 ± 0.02
pH	5.13 ± 0.06	5.28 ± 0.05

* *p* < 0.05.

**Table 4 foods-14-00962-t004:** Content of trans-isomers of fatty acids.

Indicator	Emulsion Gel (Experiment 4)	Animal Fat					
		Tail Fat	Chicken Fat	Horse Fat	Lamb Fat	Beef Fat	Pork Fat
**Trans-isomers fatty acids, from fat content in product, %**	0.179	2.908	0.14	0.412	7.239	5.024	0.26

**Table 5 foods-14-00962-t005:** Fatty acid composition of emulsion gel and different animal fats.

Name of Fatty Acid	Name of Samples						
	Emulsion Gel (Experiment 4)	Tail Fat	Chicken Fat	Horse Fat	Lamb Fat	Beef Fat	Pork Fat
Sum of saturated fatty acids, %, including:	10.85	50.42	41.22	30.23	56.24	45.04	36.33
butyric	nd	nd	nd	0.07	nd	nd	nd
capron	nd	nd	0.03	0.06	0.52	nd	nd
caprylic	nd	nd	0.04	0.04	0.02	nd	0.01
caprine	nd	0.06	0.21	0.11	0.44	0.11	0.08
undecanoic	nd	nd	0.02	nd	nd	0.06	0.01
lauric	nd	0.20	0.71	nd	0.65	0.12	0.07
tridecane	nd	0.04	0.04	0.01	0.05	0.14	nd
myristic acid	0.02	6.02	5.60	1.03	7.52	3.00	1.34
pentadecane	nd	0.73	nd	0.18	nd	2.17	0.06
palmitic	6.81	26.54	29.25	21.07	25.26	22.26	22.68
margarine	0.03	1.56	0.46	0.26	1.56	3.67	0.50
stearic	3.74	14.78	4.02	6.71	20.20	13.27	9.93
arachinic	0.19	0.35	0.35	0.42	nd	0.20	1.20
genecosan	0.05	0.09	0.29	0.27	nd	nd	0.36
behenic	0.02	0.04	0.21	nd	0.02	0.04	0.08
Sum of monounsaturated fatty acids, %, including:	17.98	47.02	43.53	40.10	38.87	45.62	56.35
(cis-9) myristoleic acid	nd	1.22	0.52	0.16	0.76	0.23	0.03
(cis-9) palmitoleic acid	0.10	2.93	7.15	3.85	1.52	4.43	3.39
(cis-10) margarinoleic acid	nd	0.74	0.62	0.46	nd	nd	0.51
(cis-9) oleic	17.73	37.90	26.99	33.54	35.68	39.72	51.88
octadecenoic	0.05	3.07	0.14	0.42	nd	nd	0.25
(cis-11) eicosenoic acid	0.09	1.16	8.11	1.68	0.91	1.25	0.30
Sum of polyunsaturated fatty acids, %, including:	71.17	2.56	15.25	29.67	4.89	9.34	7.32
linoleidic	0.13	2.31	nd	nd	nd	1.60	0.01
linoleic	70.19	nd	15.09	29.13	2.76	5.45	6.69
linolenic	0.27	0.14	0.03	0.12	0.24	0.17	0.28
eicosadiene	nd	nd	nd	nd	1.63	2.08	nd
arachidonic	nd	0.04	0.03	0.10	nd	nd	0.05
(cis-8,11,14) eicosatrienoic acid	0.56	nd	nd	nd	0.17	nd	0.02
(cis-11,14,17) eicosatrienoic	0.02	nd	0.04	0.04	0.06	0.01	0.05
eicosapentaenoic	nd	0.03	0.06	0.28	0.02	0.03	0.22
Total	100	100	100	100	100	100	100

nd—not detected.

## Data Availability

The original contributions presented in this study are included in the article. Further inquiries can be directed to the corresponding author.

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
