# Peer review of "Development and Characterization of Emulsion Gels with Pine Nut Oil, Inulin, and Whey Proteins for Reduced-Fat Meat Products"

_foods, 2025, doi:10.3390/foods14060962_

Round 1

Reviewer 1 Report

Comments and Suggestions for Authors

The manuscript discusses the potential application of emulsion gels prepared with pine nut and sunflower oils, inulin, carrageenan, and whey protein concentrate (WPC) as fat replacer in meat products. The topic of the work should be of interest to the readership of the journal. However, the manuscript requires improvement by inclusion of adequate references in some sections. Additional information should also be provided in the methodology section and the existing errors should be corrected.

I have the following comments to the authors:

  1. The authors discussed and emphasized the application of the constructed emulsion gels for replacing the animal fat in meat products in the title and throughout the manuscript. Although they did not test the application in a designed meat product. Please justify the reason for that. Emulsion gels can also be used as fat replacers in dairy products, e.g. cheese or creams. It would be instructional to the readers if the authors also provide a brief discussion in the manuscript on the potential applications of the designed gels as fat replacers in other food products.
  2. In line 198, the equation should be numbered. Please present the equation as:

Total carbohydrate content= 100-(protein+lipids+ash+moisture) %

  1. In table 2 please also include WPC and its content for different experiments 1 to 6. The impact of WPC addition in each experiment is currently unclear to the reader.
  2. In table 4, the heading should be corrected in English language.
  3. Provide an adequate reference for the information provided in lines 463-467 and 495-497.
  4. In line 67, “creating a emulsion gel” is grammatically inaccurate and should be corrected.

Please check for similar grammatical errors throughout the manuscript and correct if required.

Comments on the Quality of English Language

Minor existing grammatical errors should be corrected (please see my comments).

Author Response

Dear Reviewer

Please find attached answers

All changes are in blue

Thanks

Reviewer 2 Report

Comments and Suggestions for Authors

The article is good thoughtful. But poorly written and must be entirely revised. 

Comments on the Quality of English Language

Needs to revised.

Author Response

(The authors gave the same response as above.)

Reviewer 3 Report

Comments and Suggestions for Authors
  1. The author should appropriately introduce Pine Nut Oil in the preface, The significance of applying Inulin and Whey Proteins in research.
  2. The author needs to explain the principle and mechanism of moisturizing properties based on the characteristics of inulin.
  3. Suggest the author to modify the format of all figures according to the requirements of the journal, and refer to articles published in the journal for confirmation.
  4. The author found that 20% of inulin can be used for product development. The author needs to analyze the structural characteristics between the characteristics of inulin and the gel characteristics of the research object.
  5. The author needs to explain the possible mechanism of the interaction between inulin and carrageenan in product development and application.
  6. Suggest the author to elaborate on the principle of the interaction between inulin and carrageenan to improve the hardness of the sample.
  7. In the discussion, it is suggested to increase the benefits of research and development products for the human body.
  8. The format of references needs to be standardized, and it is recommended to modify them according to the journal format.
Comments on the Quality of English Language

None

Author Response

(The authors gave the same response as above.)

Reviewer 4 Report

Comments and Suggestions for Authors

The manuscript provides a clear and well-structured investigation of an innovative approach to reducing saturated fats in meat products through the development of emulsion gels using pine nut oil, inulin, carrageenan, and whey protein concentrate. The study is timely and relevant, addressing a gap in the current research regarding plant-derived emulsion gels in meat product formulations. The language of the article is clear and easily understandable. Text layout is preserved in accordance with the requirements of the editorial.

However, some points need to be addressed:

  1. The authors should highlight why pine nut oil, specifically, was chosen, beyond being a resource abundant in polyunsaturated fatty acids in Eastern Kazakhstan. A brief comparison to other oils could help solidify its selection.
  2. The objectives of the study are described in the last paragraph of the introduction, but they can be clarified further and expanded upon to emphasize the significance of the research. The goal should clearly state the specific improvements you're targeting.
  3. Table 4. The text should be checked and translated to English. Overall, the manuscript should be checked for typos and grammatical errors. Abbreviations should be defined just the first time they are mentioned and used as such throughout the manuscript.
  4. Table 5 caption should be better defined. Upper and lower case should be uniform. What is the difference in bold and non-bold values?
  5. The discussion of the results should be improved by comparing them with the similar findings previously published in order to highlight the significance and originality of this study.
  6. Conclusion: What is the future perspective?

Author Response

(The authors gave the same response as above.)

Round 2

Reviewer 2 Report

Comments and Suggestions for Authors

All the comments have been incorporated.

Reviewer 3 Report

Comments and Suggestions for Authors

The article can be considered acceptable.